# Supervised Word Mover's Distance

**Gao Huang**[*], **Chuan Guo**[*]
Cornell University
{gh349,cg563}@cornell.edu

**Matt J. Kusner**[†]
Alan Turing Institute, University of Warwick
mkusner@turing.ac.uk

**Yu Sun, Kilian Q. Weinberger**
Cornell University
{ys646,kqw4}@cornell.edu

**Fei Sha**
University of California, Los Angeles
feisha@cs.ucla.edu

## Abstract

Recently, a new document metric called the word mover's distance (WMD) has been proposed with unprecedented results on $k$NN-based document classification. The WMD elevates high-quality word embeddings to a document metric by formulating the distance between two documents as an optimal transport problem between the embedded words. However, the document distances are entirely unsupervised and lack a mechanism to incorporate supervision when available. In this paper we propose an efficient technique to learn a supervised metric, which we call the *Supervised-WMD (S-WMD)* metric. The supervised training minimizes the stochastic leave-one-out nearest neighbor classification error on a per-document level by updating an affine transformation of the underlying word embedding space and a word-imporance weight vector. As the gradient of the original WMD distance would result in an inefficient nested optimization problem, we provide an arbitrarily close approximation that results in a practical and efficient update rule. We evaluate S-WMD on eight real-world text classification tasks on which it consistently outperforms almost all of our *26* competitive baselines.

## 1   Introduction

Document distances are a key component of many text retrieval tasks such as web-search ranking [24], book recommendation [16], and news categorization [25]. Because of the variety of potential applications, there has been a wealth of work towards developing accurate document distances [2, 4, 11, 27]. In large part, prior work focused on extracting meaningful document representations, starting with the classical bag of words (BOW) and term frequency-inverse document frequency (TF-IDF) representations [30]. These sparse, high-dimensional representations are frequently nearly orthogonal [17] and a pair of similar documents may therefore have nearly the same distance as a pair that are very different. It is possible to design more meaningful representations through eigen-decomposing the BOW space with Latent Semantic Indexing (LSI) [11], or learning a probabilistic clustering of BOW vectors with Latent Dirichlet Allocation (LDA) [2]. Other work generalizes LDA [27] or uses denoising autoencoders [4] to learn a suitable document representation.

Recently, Kusner et al. [19] proposed the Word Mover's Distance (WMD), a new distance for text documents that leverages word embeddings [22]. Given these high-quality embeddings, the WMD defines the distances between two documents as the optimal transport cost of moving all words from one document to another within the word embedding space. This approach was shown to lead to state-of-the-art error rates in $k$-nearest neighbor ($k$NN) document classification.

---

[*]Authors contributing equally

[†]This work was done while the author was a student at Washington University in St. Louis

Importantly, these prior works are entirely *unsupervised* and not learned explicitly for any particular task. For example, text documents could be classified by *topic* or by *author*, which would lead to very different measures of dissimilarity. Lately, there has been a vast amount of work on metric learning [10, 15, 36, 37], most of which focuses on learning a generalized linear Euclidean metric. These methods often scale quadratically with the input dimensionality, and can only be applied to high-dimensional text documents after dimensionality reduction techniques such as PCA [36].

In this paper we propose an algorithm for learning a metric to improve the Word Mover's Distance. WMD stands out from prior work in that it computes distances between documents without ever learning a new document representation. Instead, it leverages low-dimensional word representations, for example word2vec, to compute distances. This allows us to transform the word embedding instead of the documents, and remain in a low-dimensional space throughout. At the same time we propose to learn word-specific 'importance' weights, to emphasize the usefulness of certain words for distinguishing the document class.

At first glance, incorporating supervision into the WMD appears computationally prohibitive, as each individual WMD computation scales cubically with respect to the (sparse) dimensionality of the documents. However, we devise an efficient technique that exploits a relaxed version of the underlying optimal transport problem, called the Sinkhorn distance [6]. This, combined with a probabilistic filtering of the training set, reduces the computation time significantly.

Our metric learning algorithm, *Supervised Word Mover's Distance (S-WMD)*, directly minimizes a stochastic version of the leave-one-out classification error under the WMD metric. Different from classic metric learning, we learn a linear transformation of the *word representations* while also learning re-weighted word frequencies. These transformations are learned to make the WMD distances match the semantic meaning of similarity encoded in the labels. We show across $8$ datasets and $26$ baseline methods the superiority of our method.

## 2  Background

Here we describe the word embedding technique we use (word2vec) and the recently introduced Word Mover's Distance. We then detail the setting of linear metric learning and the solution proposed by Neighborhood Components Analysis (NCA) [15], which inspires our method.

**word2vec** may be the most popular technique for learning a word embedding over billions of words and was introduced by Mikolov et al. [22]. Each word in the training corpus is associated with an initial word vector, which is then optimized so that if two words $w_1$ and $w_2$ frequently occur together, they have high conditional probability $p(w_2|w_1)$. This probability is the hierarchical softmax of the word vectors $\mathbf{v}_{w_1}$ and $\mathbf{v}_{w_2}$ [22], an easily-computed quantity which allows a simplified neural language model (the word2vec model) to be trained efficiently on desktop computers. Training an embedding over billions of words allows word2vec to capture surprisingly accurate word relationships [23]. Word embeddings can learn hundreds of millions of parameters and are typically by design unsupervised, allowing them to be trained on large unlabeled text corpora ahead of time. Throughout this paper we use word2vec, although many word embeddings could be used [5, 21**?** ].

**Word Mover's Distance.** Leveraging the compelling word vector relationships of word embeddings, Kusner et al. [19] introduced the *Word Mover's Distance* (WMD) as a distance between text documents. At a high level, the WMD is the minimum distance required to transport the words from one document to another. We assume that we are given a word embedding matrix $\mathbf{X} \in \mathbb{R}^{d \times n}$ for a vocabulary of $n$ words. Let $\mathbf{x}_i \in \mathcal{R}^d$ be the representation of the $i^{th}$ word, as defined by this embedding. Additionally, let $\mathbf{d}^a, \mathbf{d}^b$ be the $n$-dimensional normalized bag-of-words (BOW) vectors for two documents, where $d_i^a$ is the number of times word $i$ occurs in $\mathbf{d}^a$ (normalized over all words in $\mathbf{d}^a$). The WMD introduces an auxiliary 'transport' matrix $\mathbf{T} \in \mathcal{R}^{n \times n}$, such that $\mathbf{T}_{ij}$ describes how much of $d_i^a$ should be transported to $d_j^b$. Formally, the WMD learns $\mathbf{T}$ to minimize

$$D(\mathbf{x}_i, \mathbf{x}_j) = \min_{\mathbf{T} \geq 0} \sum_{i,j=1}^{n} \mathbf{T}_{ij} \|\mathbf{x}_i - \mathbf{x}_j\|_2^p, \quad \text{subject to,} \quad \sum_{j=1}^{n} \mathbf{T}_{ij} = d_i^a, \quad \sum_{i=1}^{n} \mathbf{T}_{ij} = d_j^b \ \forall i, j, \quad (1)$$

where $p$ is usually set to 1 or 2. In this way, documents that share many words (or even related ones) should have smaller distances than documents with very dissimilar words. It was noted in Kusner et al. [19] that the WMD is a special case of the Earth Mover's Distance (EMD) [29], also known more generally as the Wasserstein distance [20]. The authors also introduce the *word centroid distance* (WCD), which uses a fast approximation first described by Rubner et al. [29]: $\|\mathbf{X}\mathbf{d} - \mathbf{X}\mathbf{d}'\|_2$.

It can be shown that the WCD always lower bounds the WMD. Intuitively the WCD represents each document by the weighted average word vector, where the weights are the normalized BOW counts. The time complexity of solving the WMD optimization problem is $O(q^3 \log q)$ [26], where $q$ is the maximum number of unique words in either $\mathbf{d}$ or $\mathbf{d}'$. The WCD scales asymptotically by $O(dq)$.

**Regularized Transport Problem.** To alleviate the cubic time complexity of the Wasserstein distance computation, Cuturi [6] formulated a smoothed version of the underlying transport problem by adding an entropy regularizer to the transport objective. This makes the objective function strictly convex, and efficient algorithms can be adopted to solve it. In particular, given a transport matrix $\mathbf{T}$, let $h(\mathbf{T}) = -\sum_{i,j=1}^{n} \mathbf{T}_{ij} \log(\mathbf{T}_{ij})$ be the entropy of $\mathbf{T}$. For any $\lambda > 0$, the regularized (primal) transport problem is defined as

$$\min_{\mathbf{T} \geq 0} \sum_{i,j=1}^{n} \mathbf{T}_{ij} \|\mathbf{x}_i - \mathbf{x}_j\|_2^p - \frac{1}{\lambda} h(\mathbf{T}) \quad \text{subject to,} \quad \sum_{j=1}^{n} \mathbf{T}_{ij} = d_i^a, \ \sum_{i=1}^{n} \mathbf{T}_{ij} = d_j^b \ \forall i,j. \quad (2)$$

The larger $\lambda$ is, the closer this relaxation is to the original Wasserstein distance. Cuturi [6] propose an efficient algorithm to solve for the optimal transport $\mathbf{T}_\lambda^*$ using a clever matrix-scaling algorithm. Specifically, we may define the matrix $\mathbf{K}_{ij} = \exp(-\lambda \|\mathbf{x}_i - \mathbf{x}_j\|_2)$ and solve for the scaling vectors $\mathbf{u}, \mathbf{v}$ to a fixed-point by computing $\mathbf{u} = \mathbf{d}^a ./ (\mathbf{K}\mathbf{v})$, $\mathbf{v} = \mathbf{d}^b ./ (\mathbf{K}^\top \mathbf{u})$ in an alternating fashion. These yield the relaxed transport $\mathbf{T}_\lambda^* = \operatorname{diag}(\mathbf{u}) \mathbf{K} \operatorname{diag}(\mathbf{v})$. This algorithm can be shown to have empirical time complexity $O(q^2)$ [6], which is significantly faster than solving the WMD problem exactly.

**Linear Metric Learning.** Assume that we have access to a training set $\{\mathbf{x}_1, \ldots, \mathbf{x}_n\} \subset \mathbb{R}^d$, arranged as columns in matrix $\mathbf{X} \in \mathbb{R}^{d \times n}$, and corresponding labels $\{y_1, \ldots, y_n\} \subseteq \mathcal{Y}^n$, where $\mathcal{Y}$ contains some finite number of classes $C = |\mathcal{Y}|$. Linear metric learning learns a matrix $\mathbf{A} \in \mathbb{R}^{r \times d}$, where $r \leq d$, and defines the generalized Euclidean distance between two documents $\mathbf{x}_i$ and $\mathbf{x}_j$ as $d_{\mathbf{A}}(\mathbf{x}_i, \mathbf{x}_j) = \|\mathbf{A}(\mathbf{x}_i - \mathbf{x}_j)\|_2$. Popular linear metric learning algorithms are NCA [15], LMNN [36], and ITML [10] amongst others [37]. These methods learn a matrix $\mathbf{A}$ to minimize a loss function that is often an approximation of the leave-one-out (LOO) classification error of the $k$NN classifier.

**Neighborhood Components Analysis (NCA)** was introduced by Goldberger et al. [15] to learn a generalized Euclidean metric. Here, the authors approximate the non-continuous leave-one-out $k$NN error by defining a stochastic neighborhood process. An input $\mathbf{x}_i$ is assigned input $\mathbf{x}_j$ as its nearest neighbor with probability

$$p_{ij} = \frac{\exp(-d_{\mathbf{A}}^2(\mathbf{x}_i, \mathbf{x}_j))}{\sum_{k \neq i} \exp(-d_{\mathbf{A}}^2(\mathbf{x}_i, \mathbf{x}_k))}, \quad (3)$$

where we define $p_{ii} = 0$. Under this stochastic neighborhood assignment, an input $\mathbf{x}_i$ with label $y_i$ is classified correctly if its nearest neighbor is any $\mathbf{x}_j \neq \mathbf{x}_i$ from the same class ($y_j = y_i$). The probability of this event can be stated as $p_i = \sum_{j:y_j=y_i} p_{ij}$. NCA learns $\mathbf{A}$ by maximizing the expected LOO accuracy $\sum_i p_i$, or equivalently by minimizing $-\sum_i \log(p_i)$, the KL-divergence from a perfect classification distribution ($p_i = 1$ for all $\mathbf{x}_i$).

## 3  Learning a Word Embedding Metric

In this section we propose a method for learning a supervised document distance, by way of learning a generalized Euclidean metric within the word embedding space and a word importance vector. We will refer to the learned document distance as the *Supervised Word Mover's Distance (S-WMD)*. To learn such a metric we assume we have a training dataset consisting of $m$ documents $\{\mathbf{d}^1, \ldots, \mathbf{d}^m\} \subset \Sigma_n$, where $\Sigma_n$ is the $(n-1)$-dimensional simplex (thus each document is represented as a normalized histogram over the words in the vocabulary, of size $n$). For each document we are given a label out of $C$ possible classes, *i.e.* $\{y_1, \ldots, y_m\} \subseteq \{1, \ldots, C\}^m$. Additionally, we are given a word embedding matrix $\mathbf{X} \in \mathbb{R}^{d \times n}$ (e.g., the word2vec embedding) which defines a $d$-dimensional word vector for each of the words in the vocabulary.

**Supervised WMD.** As described in the previous section, it is possible to define a distance between any two documents $\mathbf{d}^a$ and $\mathbf{d}^b$ as the minimum cumulative word distance of moving $\mathbf{d}^a$ to $\mathbf{d}^b$ in word embedding space, as is done in the WMD. Given a labeled training set we would like to improve the distance so that documents that share the same labels are close, and those with different labels are far apart. We capture this notion of similarity in two ways: First we transform the word embedding, which captures a latent representation of words. We adapt this representation with a

linear transformation $\mathbf{x}_i \rightarrow \mathbf{A}\mathbf{x}_i$, where $\mathbf{x}_i$ represents the embedding of the $i^{th}$ word. Second, as different classification tasks and data sets may value words differently, we also introduce a histogram importance vector $\mathbf{w}$ that re-weighs the word histogram values to reflect the importance of words for distinguishing the classes:

$$\tilde{\mathbf{d}}^a = (\mathbf{w} \circ \mathbf{d}^a)/(\mathbf{w}^\top \mathbf{d}^a), \tag{4}$$

where "$\circ$" denotes the element-wise Hadamard product. After applying the vector $\mathbf{w}$ and the linear mapping $\mathbf{A}$, the WMD distance between documents $\mathbf{d}^a$ and $\mathbf{d}^b$ becomes

$$D_{\mathbf{A},\mathbf{w}}(\mathbf{d}^a, \mathbf{d}^b) \triangleq \min_{\mathbf{T} \geq 0} \sum_{i,j=1}^{n} \mathbf{T}_{ij} \|\mathbf{A}(\mathbf{x}_i - \mathbf{x}_j)\|_2^2 \text{ s.t. } \sum_{j=1}^{n} \mathbf{T}_{ij} = \tilde{d}_i^a \text{ and } \sum_{i=1}^{n} \mathbf{T}_{ij} = \tilde{d}_j^b \quad \forall i, j. \tag{5}$$

**Loss Function.** Our goal is to learn the matrix $\mathbf{A}$ and vector $\mathbf{w}$ to make the distance $D_{\mathbf{A},\mathbf{w}}$ reflect the semantic definition of similarity encoded in the labeled data. Similar to prior work on metric learning [10, 15, 36] we achieve this by minimizing the $k$NN-LOO error with the distance $D_{\mathbf{A},\mathbf{w}}$ in the document space. As the LOO error is non-differentiable, we use the stochastic neighborhood relaxation proposed by Hinton & Roweis [18], which is also used for NCA. Similar to prior work we use the squared Euclidean word distance in Eq. (5). We use the KL-divergence loss proposed in NCA alongside the definition of neighborhood probability in (3) which yields,

$$\ell(\mathbf{A}, \mathbf{w}) = -\sum_{a=1}^{m} \log \left( \sum_{b:y_b = y_a}^{m} \frac{\exp(-D_{\mathbf{A},\mathbf{w}}(\mathbf{d}_a, \mathbf{d}_b))}{\sum_{c \neq a} \exp\left(-D_{\mathbf{A},\mathbf{w}}(\mathbf{d}_a, \mathbf{d}_c)\right)} \right). \tag{6}$$

**Gradient.** We can compute the gradient of the loss $\ell(\mathbf{A}, \mathbf{w})$ with respect to $\mathbf{A}$ and $\mathbf{w}$ as follows,

$$\frac{\partial}{\partial(\mathbf{A}, \mathbf{w})} \ell(\mathbf{A}, \mathbf{w}) = \sum_{a=1}^{m} \sum_{b \neq a} \frac{p_{ab}}{p_a} (\delta_{ab} - p_a) \frac{\partial}{\partial(\mathbf{A}, \mathbf{w})} D_{\mathbf{A},\mathbf{w}}(\mathbf{d}^a, \mathbf{d}^b), \tag{7}$$

where $\delta_{ab} = 1$ if and only if $y_a = y_b$, and $\delta_{ab} = 0$ otherwise.

## 3.1 Fast computation of $\partial D_{\mathbf{A},\mathbf{w}}(\mathbf{d}^a, \mathbf{d}^b)/\partial(\mathbf{A}, \mathbf{w})$

Notice that the remaining gradient term above $\partial D_{\mathbf{A},\mathbf{w}}(\mathbf{d}^a, \mathbf{d}^b)/\partial(\mathbf{A}, \mathbf{w})$ contains the nested linear program defined in (5). In fact, computing this gradient just for *a single pair of documents* will require time complexity $O(q^3 \log q)$, where $q$ is the largest set of unique words in either document [8]. This quickly becomes prohibitively slow as the document size becomes large and the number of documents increase. Further, the gradient is not always guaranteed to exist [1, 7] (instead we must resort to subgradient descent). Motivated by the recent works on fast Wasserstein distance computation [6, 8, 12], we propose to relax the modified linear program in eq. (5) using the entropy as in eq. (2). As described in Section 2, this allows us to approximately solve eq. (5) in $O(q^2)$ time via $\mathbf{T}_\lambda^* = \text{diag}(\mathbf{u})\mathbf{K}\,\text{diag}(\mathbf{v})$. We will use this approximate solution in the following gradients.

**Gradient w.r.t. A.** It can be shown that,

$$\frac{\partial}{\partial \mathbf{A}} D_{\mathbf{A},\mathbf{w}}(\mathbf{d}^a, \mathbf{d}^b) = 2\mathbf{A} \sum_{i,j=1}^{n} \mathbf{T}_{ij}^{ab}(\mathbf{x}_i - \mathbf{x}_j)(\mathbf{x}_i - \mathbf{x}_j)^\top, \tag{8}$$

where $\mathbf{T}^{ab}$ is the optimizer of (5), so long as it is unique (otherwise it is a subgradient) [1]. We replace $\mathbf{T}^{ab}$ by $\mathbf{T}_\lambda^*$ which is always unique as the relaxed transport is strongly convex [9].

**Gradient w.r.t. w.** To obtain the gradient with respect to $\mathbf{w}$, we need the optimal solution to the dual transport problem:

$$D_{\mathbf{A},\mathbf{w}}^*(\mathbf{d}^a, \mathbf{d}^b) \triangleq \max_{(\boldsymbol{\alpha}, \boldsymbol{\beta})} \boldsymbol{\alpha}^\top \tilde{\mathbf{d}}^a + \boldsymbol{\beta}^\top \tilde{\mathbf{d}}^b; \text{ s.t. } \alpha_i + \beta_j \leq \|\mathbf{A}(\mathbf{x}_i - \mathbf{x}_j)\|_2^2 \quad \forall i, j. \tag{9}$$

Given that both $\tilde{\mathbf{d}}^a$ and $\tilde{\mathbf{d}}^b$ are functions of $\mathbf{w}$, we have

$$\frac{\partial}{\partial \mathbf{w}} D_{\mathbf{A},\mathbf{w}}(\mathbf{d}^a, \mathbf{d}^b) = \frac{\partial D_{\mathbf{A},\mathbf{w}}^*}{\partial \tilde{\mathbf{d}}^a} \frac{\partial \tilde{\mathbf{d}}^a}{\partial \mathbf{w}} + \frac{\partial D_{\mathbf{A},\mathbf{w}}^*}{\partial \tilde{\mathbf{d}}^b} \frac{\partial \tilde{\mathbf{d}}^b}{\partial \mathbf{w}} = \frac{\boldsymbol{\alpha}^* \circ \mathbf{d}^a - (\boldsymbol{\alpha}^{*\top} \tilde{\mathbf{d}}^a)\mathbf{d}^a}{\mathbf{w}^\top \mathbf{d}^a} + \frac{\boldsymbol{\beta}^* \circ \mathbf{d}^b - (\boldsymbol{\beta}^{*\top} \tilde{\mathbf{d}}^b)\mathbf{d}^b}{\mathbf{w}^\top \mathbf{d}^b}. \tag{10}$$

Instead of solving the dual directly, we obtain the relaxed optimal dual variables $\boldsymbol{\alpha}_\lambda^*, \boldsymbol{\beta}_\lambda^*$ via the vectors $\mathbf{u}, \mathbf{v}$ that were used to derive our relaxed transport $\mathbf{T}_\lambda^*$. Specifically, we can solve for the dual variables as such: $\boldsymbol{\alpha}_\lambda^* = \frac{\log(\mathbf{u})}{\lambda} - \frac{\log(\mathbf{u})^\top \mathbf{1}}{p}\mathbf{1}$ and $\boldsymbol{\beta}_\lambda^* = \frac{\log(\mathbf{v})}{\lambda} - \frac{\log(\mathbf{v})^\top \mathbf{1}}{p}\mathbf{1}$, where $\mathbf{1}$ is the $p$-dimensional all ones vector. In general, we can observe from eq. (2) that the above approximation process becomes more accurate as $\lambda$ grows. However, setting $\lambda$ too large can make the algorithm converges slower. In our experiments, we use $\lambda = 10$, which leads to a nice trade-off between speed and approximation accuracy.

## 3.2 Optimization

Alongside the fast gradient computation process introduced above, we can further speed up the training with a clever initialization and batch gradient descent.

**Initialization.** The loss function in eq. (6) is nonconvex and is thus highly dependent on the initial setting of $\mathbf{A}$ and $\mathbf{w}$. A good initialization also drastically reduces the number of gradient steps required. For $\mathbf{w}$, we initialize all its entries to 1, i.e., all words are assigned with the same weights at the beginning. For $\mathbf{A}$, we propose to learn an initial projection within the word centroid distance (WCD), defined as $D'(\mathbf{d}^a, \mathbf{d}^b) = \|\mathbf{X}\mathbf{d}^a - \mathbf{X}\mathbf{d}^b\|_2$, described in Sec-

---
**Algorithm 1** S-WMD

1: **Input:** word embedding: $\mathbf{X}$,
2: dataset: $\{(\mathbf{d}^1, y_1), \ldots, (\mathbf{d}^m, y_m)\}$
3: $\mathbf{c}^a = \mathbf{X}\mathbf{d}^a, \; \forall a \in \{1, \ldots, m\}$
4: $\mathbf{A} = \text{NCA}((\mathbf{c}^1, y_1), \ldots, (\mathbf{c}^m, y_m))$
5: $\mathbf{w} = \mathbf{1}$
6: **while** loop until convergence **do**
7:     Randomly select $\mathcal{B} \subseteq \{1, \ldots, m\}$
8:     Compute gradients using eq. (11)
9:     $\mathbf{A} \leftarrow \mathbf{A} - \eta_{\mathbf{A}}\mathbf{g}_{\mathbf{A}}$
10:     $\mathbf{w} \leftarrow \mathbf{w} - \eta_{\mathbf{w}}\mathbf{g}_{\mathbf{w}}$
11: **end while**

---

tion 2. The WCD should be a reasonable approximation to the WMD. Kusner et al. [19] point out that the WCD is a lower bound on the WMD, which holds true after the transformation with $\mathbf{A}$. We obtain our initialization by applying NCA in word embedding space using the WCD distance between documents. This is to say that we can construct the WCD dataset: $\{\mathbf{c}^1, \ldots, \mathbf{c}^m\} \subset \mathbb{R}^d$, representing each text document as its word centroid, and apply NCA in the usual way as described in Section 2. We call this learned word distance *Supervised Word Centroid Distance (S-WCD)*.

**Batch Gradient Descent.** Once the initial matrix $\mathbf{A}$ is obtained, we minimize the loss $\ell(\mathbf{A}, \mathbf{w})$ in (6) with batch gradient descent. At each iteration, instead of optimizing over the full training set, we randomly pick a batch of documents $\mathcal{B}$ from the training set, and compute the gradient for these documents. We can further speed up training by observing that the vast majority of NCA probabilities $p_{ab}$ near zero. This is because most documents are far away from any given document. Thus, for a document $\mathbf{d}^a$ we can use the WCD to get a cheap neighbor ordering and only compute the NCA probabilities for the closest set of documents $\mathcal{N}_a$, based on the WCD. When we compute the gradient for each of the selected documents, we only use the document's $M$ nearest neighbor documents (defined by WCD distance) to compute the NCA neighborhood probabilities. In particular, the gradient is computed as follows,

$$\mathbf{g}_{\mathbf{A},\mathbf{w}} = \sum_{a \in \mathcal{B}} \sum_{b \in \mathcal{N}_a} (p_{ab}/p_a)(\delta_{ab} - p_a) \frac{\partial}{\partial(\mathbf{A}, \mathbf{w})} D_{(\mathbf{A},\mathbf{w})}(\mathbf{d}^a, \mathbf{d}^b), \tag{11}$$

where again $\mathcal{N}_a$ is the set of nearest neighbors of document $a$. With the gradient, we update $\mathbf{A}$ and $\mathbf{w}$ with learning rates $\eta_{\mathbf{A}}$ and $\eta_{\mathbf{w}}$, respectively. Algorithm 1 summarizes S-WMD in pseudo code.

**Complexity.** The empirical time complexity of solving the dual transport problem scales quadratically with $p$ [26]. Therefore, the complexity of our algorithm is $O(TBN[p^2 + d^2(p + r)])$, where $T$ denotes the number of batch gradient descent iterations, $B = |\mathcal{B}|$ the batch size, $N = |\mathcal{N}_a|$ the size of the nearest neighbor set, and $p$ the maximum number of unique words in a document. This is because computing $\mathbf{T}_{ij}^*$, $\boldsymbol{\alpha}^*$ and $\boldsymbol{\beta}^*$ using the alternating fixed point algorithm in Section 3.1 requires $O(p^2)$ time, while constructing the gradients from eqs. (8) and (10) takes $O(d^2(p + r))$ time. The approximated gradient eq. (11) requires this computation to be repeated $BN$ times. In our experiments, we set $B = 32$ and $N = 200$, and computing the gradient at each iteration can be done in seconds.

## 4 Results

We evaluate S-WMD on 8 different document corpora and compare the $k$NN error with unsupervised WCD, WMD, and 6 document representations. In addition, all 6 document representation baselines

Table 1: The document datasets (and their descriptions) used for visualization and evaluation.

| name | description | $C$ | $n$ | $ne$ | BOW dim. | avg words |
|---|---|---|---|---|---|---|
| BBCSPORT | BBC sports articles labeled by sport | 5 | 517 | 220 | 13243 | 117 |
| TWITTER | tweets categorized by sentiment [31] | 3 | 2176 | 932 | 6344 | 9.9 |
| RECIPE | recipe procedures labeled by origin | 15 | 3059 | 1311 | 5708 | 48.5 |
| OHSUMED | medical abstracts (class subsampled) | 10 | 3999 | 5153 | 31789 | 59.2 |
| CLASSIC | academic papers labeled by publisher | 4 | 4965 | 2128 | 24277 | 38.6 |
| REUTERS | news dataset (train/test split [3]) | 8 | 5485 | 2189 | 22425 | 37.1 |
| AMAZON | reviews labeled by product | 4 | 5600 | 2400 | 42063 | 45.0 |
| 20NEWS | canonical news article dataset [3] | 20 | 11293 | 7528 | 29671 | 72 |

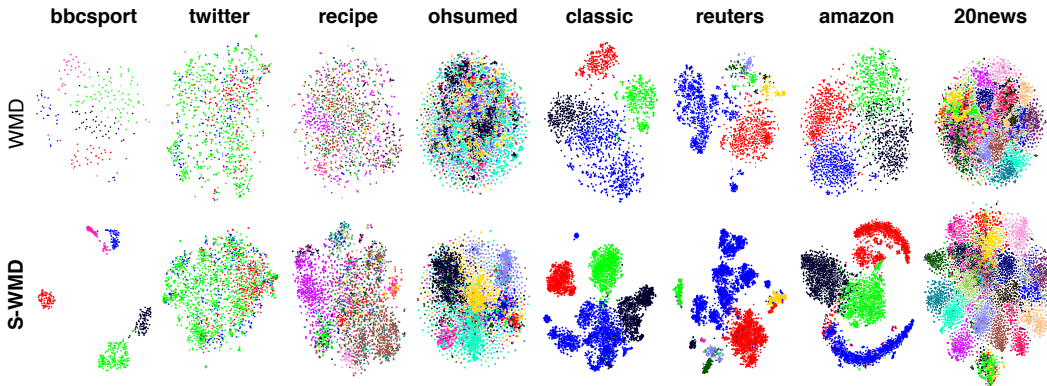

Figure 1: t-SNE plots of WMD and S-WMD on all datasets.

are used with and without 3 leading supervised metric learning algorithms—resulting in an overall total of 26 competitive baselines. Our code is implemented in Matlab and is freely available at `https://github.com/gaohuang/S-WMD`.

**Datasets and Baselines.** We evaluate all approaches on 8 document datasets in the settings of news categorization, sentiment analysis, and product identification, among others. Table 1 describes the classification tasks as well as the size and number of classes $C$ of each of the datasets. We evaluate against the following document representation/distance methods: 1. *bag-of-words* (BOW): a count of the number of word occurrences in a document, the length of the vector is the number of unique words in the corpus; 2. *term frequency-inverse document frequency* (TF-IDF): the BOW vector normalized by the document frequency of each word across the corpus; 3. *Okapi BM25* [28]: a TF-IDF-like ranking function, first used in search engines; 4. *Latent Semantic Indexing* (LSI) [11]: projects the BOW vectors onto an orthogonal basis via singular value decomposition; 5. *Latent Dirichlet Allocation* (LDA) [2]: a generative probabilistic method that models documents as mixtures of word 'topics'. We train LDA *transductively* (i.e., on the combined collection of training & testing words) and use the topic probabilities as the document representation ; 6. *Marginalized Stacked Denoising Autoencoders* (mSDA) [4]: a fast method for training stacked denoising autoencoders, which have state-of-the-art error rates on sentiment analysis tasks [14]. For datasets larger than RECIPE we use either a high-dimensional variant of mSDA or take 20% of the features that occur most often, whichever has better performance.; 7. *Word Centroid Distance* (WCD), described in Section 2; 8. *Word Mover's Distance* (WMD), described in Section 2. For completeness, we also show results for the Supervised Word Centroid Distance (S-WCD) and the initialization of S-WMD (S-WMD init.), described in Section 3. For methods that propose a document representation (as opposed to a distance), we use the Euclidean distance between these vector representations for visualization and $k$NN classification. For the supervised metric learning results we first reduce the dimensionality of each representation to 200 dimensions (if necessary) with PCA and then run either NCA, ITML, or LMNN on the projected data. We tune all free hyperparameters in all compared methods with Bayesian optimization (BO), using the implementation of Gardner et al. [13][3].

$k$**NN classification.** We show the $k$NN test error of all document representation and distance methods in Table 2. For datasets that do not have a predefined train/test split: BBCSPORT, TWITTER, RECIPE, CLASSIC, and AMAZON we average results over five 70/30 train/test splits and report standard errors. For each dataset we highlight the best results in bold (and those whose standard error

Table 2: The $k$NN test error for all datasets and distances.

| DATASET | BBCSPORT | TWITTER | RECIPE | OHSUMED | CLASSIC | REUTERS | AMAZON | 20NEWS | AVERAGE-RANK |
|---|---|---|---|---|---|---|---|---|---|
| UNSUPERVISED | | | | | | | | | |
| BOW | $20.6 \pm 1.2$ | $43.6 \pm 0.4$ | $59.3 \pm 1.0$ | 61.1 | $36.0 \pm 0.5$ | 13.9 | $28.5 \pm 0.5$ | 57.8 | 26.1 |
| TF-IDF | $21.5 \pm 2.8$ | $33.2 \pm 0.9$ | $53.4 \pm 1.0$ | 62.7 | $35.0 \pm 1.8$ | 29.1 | $41.5 \pm 1.2$ | 54.4 | 25.0 |
| OKAPI BM25 [28] | $16.9 \pm 1.5$ | $42.7 \pm 7.8$ | $53.4 \pm 1.9$ | 66.2 | $40.6 \pm 2.7$ | 32.8 | $58.8 \pm 2.6$ | 55.9 | 26.1 |
| LSI [11] | $4.3 \pm 0.6$ | $31.7 \pm 0.7$ | $45.4 \pm 0.5$ | 44.2 | $6.7 \pm 0.4$ | 6.3 | $9.3 \pm 0.4$ | 28.9 | 12.0 |
| LDA [2] | $6.4 \pm 0.7$ | $33.8 \pm 0.3$ | $51.3 \pm 0.6$ | 51.0 | $5.0 \pm 0.3$ | 6.9 | $11.8 \pm 0.6$ | 31.5 | 16.6 |
| MSDA [4] | $8.4 \pm 0.8$ | $32.3 \pm 0.7$ | $48.0 \pm 1.4$ | 49.3 | $6.9 \pm 0.4$ | 8.1 | $17.1 \pm 0.4$ | 39.5 | 18.0 |
| ITML [10] | | | | | | | | | |
| BOW | $7.4 \pm 1.4$ | $32.0 \pm 0.4$ | $63.1 \pm 0.9$ | 70.1 | $7.5 \pm 0.5$ | 7.3 | $20.5 \pm 2.1$ | 60.6 | 23.0 |
| TF-IDF | $1.8 \pm 0.2$ | $31.1 \pm 0.3$ | $51.0 \pm 1.4$ | 55.1 | $9.9 \pm 1.0$ | 6.6 | $11.1 \pm 1.9$ | 45.3 | 14.8 |
| OKAPI BM25 [28] | $3.7 \pm 0.5$ | $31.9 \pm 0.3$ | $53.8 \pm 1.8$ | 77.0 | $18.3 \pm 4.5$ | 20.7 | $11.4 \pm 2.9$ | 81.5 | 21.5 |
| LSI [11] | $5.0 \pm 0.7$ | $32.3 \pm 0.4$ | $55.7 \pm 0.8$ | 54.7 | $5.5 \pm 0.7$ | 6.9 | $10.6 \pm 2.2$ | 39.6 | 17.6 |
| LDA [2] | $6.5 \pm 0.7$ | $33.9 \pm 0.9$ | $59.3 \pm 0.8$ | 59.6 | $6.6 \pm 0.5$ | 9.2 | $15.7 \pm 2.0$ | 87.8 | 22.5 |
| MSDA [4] | $25.5 \pm 9.4$ | $43.7 \pm 7.4$ | $54.5 \pm 1.3$ | 61.8 | $14.9 \pm 2.2$ | 5.9 | $37.4 \pm 4.0$ | 47.7 | 23.9 |
| LMNN [36] | | | | | | | | | |
| BOW | $2.4 \pm 0.4$ | $31.8 \pm 0.3$ | $48.4 \pm 0.4$ | 49.1 | $4.7 \pm 0.3$ | 3.9 | $10.7 \pm 0.3$ | 40.7 | 11.5 |
| TF-IDF | $4.0 \pm 0.6$ | $30.8 \pm 0.3$ | $43.7 \pm 0.3$ | 40.0 | $4.9 \pm 0.3$ | 5.8 | $6.8 \pm 0.3$ | 28.1 | 7.8 |
| OKAPI BM25 [28] | $1.9 \pm 0.7$ | $30.5 \pm 0.4$ | $41.7 \pm 0.7$ | 59.4 | $19.0 \pm 9.3$ | 9.2 | $6.9 \pm 0.2$ | 57.4 | 14.4 |
| LSI [11] | $2.4 \pm 0.5$ | $31.6 \pm 0.2$ | $44.8 \pm 0.4$ | 40.8 | $3.0 \pm 0.1$ | **3.2** | $6.6 \pm 0.2$ | **25.1** | 5.1 |
| LDA [2] | $4.5 \pm 0.4$ | $31.9 \pm 0.6$ | $51.4 \pm 0.4$ | 49.9 | $4.9 \pm 0.4$ | 5.6 | $12.1 \pm 0.6$ | 32.0 | 14.6 |
| MSDA [4] | $22.7 \pm 10.0$ | $50.3 \pm 8.6$ | $46.3 \pm 1.2$ | 41.6 | $11.1 \pm 1.9$ | 5.3 | $24.0 \pm 3.6$ | 27.1 | 17.3 |
| NCA [15] | | | | | | | | | |
| BOW | $9.6 \pm 0.6$ | $31.1 \pm 0.5$ | $55.2 \pm 0.6$ | 57.4 | $4.0 \pm 0.1$ | 6.2 | $16.8 \pm 0.3$ | 46.4 | 17.5 |
| TF-IDF | $\mathbf{0.6 \pm 0.3}$ | $30.6 \pm 0.5$ | $41.4 \pm 0.4$ | 35.8 | $5.5 \pm 0.2$ | 3.8 | $6.5 \pm 0.2$ | 29.3 | 5.4 |
| OKAPI BM25 [28] | $4.5 \pm 0.5$ | $31.8 \pm 0.4$ | $45.8 \pm 0.5$ | 56.6 | $20.6 \pm 4.8$ | 10.5 | $8.5 \pm 0.4$ | 55.9 | 17.9 |
| LSI [11] | $2.4 \pm 0.7$ | $31.1 \pm 0.8$ | $41.6 \pm 0.5$ | 37.5 | $3.1 \pm 0.2$ | 3.3 | $7.7 \pm 0.4$ | 30.7 | 6.3 |
| LDA [2] | $7.1 \pm 0.9$ | $32.7 \pm 0.3$ | $50.9 \pm 0.4$ | 50.7 | $5.0 \pm 0.2$ | 7.9 | $11.6 \pm 0.8$ | 30.9 | 16.5 |
| MSDA [4] | $21.8 \pm 7.4$ | $37.9 \pm 2.8$ | $48.0 \pm 1.6$ | 40.4 | $11.2 \pm 1.8$ | 5.2 | $23.6 \pm 3.1$ | 26.8 | 16.1 |
| DISTANCES IN THE WORD MOVER'S FAMILY | | | | | | | | | |
| WCD [19] | $11.3 \pm 1.1$ | $30.7 \pm 0.9$ | $49.4 \pm 0.3$ | 48.9 | $6.6 \pm 0.2$ | 4.7 | $9.2 \pm 0.2$ | 36.2 | 13.5 |
| WMD [19] | $4.6 \pm 0.7$ | $28.7 \pm 0.6$ | $42.6 \pm 0.3$ | 44.5 | $\mathbf{2.8 \pm 0.1}$ | 3.5 | $7.4 \pm 0.3$ | 26.8 | 6.1 |
| **S-WCD** | $4.6 \pm 0.5$ | $30.4 \pm 0.5$ | $51.3 \pm 0.2$ | 43.3 | $5.8 \pm 0.2$ | 3.9 | $7.6 \pm 0.3$ | 33.6 | 11.4 |
| **S-WMD INIT.** | $2.8 \pm 0.3$ | $28.2 \pm 0.4$ | $39.8 \pm 0.4$ | 38.0 | $3.3 \pm 0.3$ | 3.5 | $\mathbf{5.8 \pm 0.2}$ | 28.4 | 4.3 |
| **S-WMD** | $2.1 \pm 0.5$ | $\mathbf{27.5 \pm 0.5}$ | $\mathbf{39.2 \pm 0.3}$ | 34.3 | $3.2 \pm 0.2$ | **3.2** | $\mathbf{5.8 \pm 0.1}$ | 26.8 | **2.4** |

overlaps the mean of the best result). On the right we also show the average rank across datasets, relative to unsupervised BOW (bold indicates the best method). We highlight the unsupervised WMD in blue (WMD) and our new result in red (S-WMD). Despite the very large number of competitive baselines, S-WMD achieves the lowest $k$NN test error on 5/8 datasets, with the exception of BBCSPORT, CLASSIC and AMAZON. On these datasets it achieves the 4th lowest on BBCSPORT and CLASSIC, and tied at 2nd on 20NEWS. On average across all datasets it outperforms all other 26 methods. Another observation is that S-WMD right after initialization (S-WMD init.) performs quite well. However, as training S-WMD is efficient (shown in Table 3), it is often well worth the training time.

For unsupervised baselines, on datasets BBCSPORT and OHSUMED, where the previous state-of-the-art WMD was beaten by LSI, S-WMD reduces the error of LSI relatively by $51\%$ and $22\%$, respectively. In general, supervision seems to help all methods on average. One reason why NCA with a TF-IDF document representation may be performing better than S-WMD could be because of the long document lengths in BBCSPORT and OHSUMED. Having denser BOW vectors may improve the inverse document frequency weights, which in turn may be a good initialization for NCA to further fine-tune. On datasets with smaller documents such as TWITTER, CLASSIC, and REUTERS, S-WMD outperforms NCA with TF-IDF relatively by $10\%$, $42\%$, and $15\%$, respectively. On CLASSIC WMD outperforms S-WMD possibly because of a poor initialization and that S-WMD uses the squared Euclidean distance between word vectors, which may be suboptimal for this dataset. This however, does not occur for any other dataset.

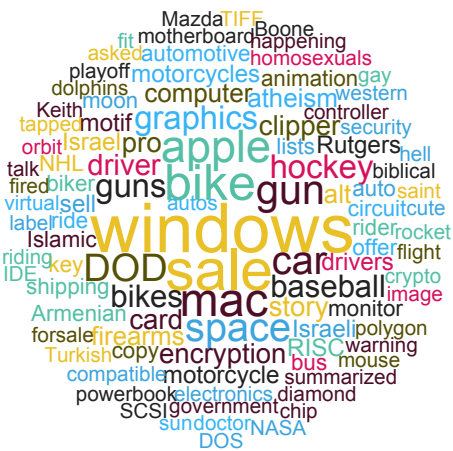

Figure 2: The Top-100 words upweighted by S-WMD on 20NEWS.

**Visualization.** Figure 1 shows a 2D embedding of the test split of each dataset by WMD and S-WMD using t-Stochastic Neighbor Embedding (t-SNE) [33]. The quality of a distance can be visualized by how clustered points in the same class are. Using this metric, S-WMD noticeably improves upon WMD on almost all the 8 datasets. Figure 2 visualizes the top 100 words with

largest weights learned by S-WMD on the 20NEWS dataset. The size of each word is proportional its learned weight. We can observe that these upweighted words are indeed most representative for the true classes of this dataset. More detailed results and analysis can be found in the supplementary.

**Training time.** Table 3 shows the training times for S-WMD. Note that the time to learn the initial metric $\mathbf{A}$ is not included in time shown in the second column. Relative to the initialization, S-WMD is surprisingly fast. This is due to the fast gradient approximation and the batch gradient descent introduced in Section 3.1 and 3.2. We note that these times are comparable or even faster than the time it takes to train a linear metric on the baseline methods after PCA.

Table 3: Distance computation times.

| FULL TRAINING TIMES | | |
|---|---|---|
| DATASET | METRICS | |
| | S-WCD/S-WMD INIT. | S-WMD |
| BBCSPORT | 1M 25S | 4M 56S |
| TWITTER | 28M 59S | 7M 53S |
| RECIPE | 23M 21S | 23M 58S |
| OHSUMED | 46M 18S | 29M 12S |
| CLASSIC | 1H 18M | 36M 22S |
| REUTERS | 2H 7M | 34M 56S |
| AMAZON | 2H 15M | 20M 10S |
| 20NEWS | 14M 42S | 1H 55M |

## 5  Related Work

Metric learning is a vast field that includes both supervised and unsupervised techniques (see Yang & Jin [37] for a large survey). Alongside NCA [15], described in Section 2, there are a number of popular methods for generalized Euclidean metric learning. Large Margin Nearest Neighbors (LMNN) [36] learns a metric that encourages inputs with similar labels to be close in a local region, while encouraging inputs with different labels to be farther by a large margin. Information-Theoretic Metric Learning (ITML) [10] learns a metric by minimizing a KL-divergence subject to generalized Euclidean distance constraints. Cuturi & Avis [7] was the first to consider learning the ground distance in the Earth Mover's Distance (EMD). In a similar work, Wang & Guibas [34] learns a ground distance that is not a metric, with good performance in certain vision tasks. Most similar to our work Wang et al. [35] learn a metric within a generalized Euclidean EMD ground distance using the framework of ITML for image classification. They do not, however, consider re-weighting the histograms, which allows our method extra flexibility. Until recently, there has been relatively little work towards learning supervised word embeddings, as state-of-the-art results rely on making use of large unlabeled text corpora. Tang et al. [32] propose a neural language model that uses label information from emoticons to learn sentiment-specific word embeddings.

## 6  Conclusion

We proposed a powerful method to learn a supervised word mover's distance, and demonstrated that it may well be the best performing distance metric for documents to date. Similar to WMD, our S-WMD benefits from the large unsupervised corpus, which was used to learn the word2vec embedding [22, 23]. The word embedding gives rise to a very good document distance, which is particularly forgiving when two documents use syntactically different but conceptually similar words. Two words may be similar in one sense but dissimilar in another, depending on the articles in which they are contained. It is these differences that S-WMD manages to capture through supervised training. By learning a linear metric and histogram re-weighting through the optimal transport of the word mover's distance, we are able to produce state-of-the-art classification results efficiently.

**Acknowledgments**

The authors are supported in part by the, III-1618134, III-1526012, IIS-1149882 grants from the National Science Foundation and the Bill and Melinda Gates Foundation. We also thank Dor Kedem for many insightful discussions.

## Footnotes

[3]`http://tinyurl.com/bayesopt`

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
