[Supplementary Material]

# Supervised Word Mover's Distance: Supplementary Material

## 1 Results of only learn **w** or **A** in S-WMD

We run experiments to analyze the independent contribution of learning the word weights **w** or distance metric **A** in S-WMD. In Table 1, S-WMD (**w** only) and S-WMD (**A** only) respectively represent the variants of S-WMD algorithm which only learn **w** and **A**. One can observe that only updating one of the two set of parameters generally improves the performance over S-WMD init, but with small margins. In comparison, S-WMD yields more significant improvement by jointly learning both **w** and **A**.

Table 1: The $k$NN test error on all datasets.

| DATASET | BBCSPORT | TWITTER | RECIPE | OHSUMED | CLASSIC | REUTERS | AMAZON | 20NEWS |
|---|---|---|---|---|---|---|---|---|
| **S-WCD** | $4.6 \pm 0.5$ | $30.4 \pm 0.5$ | $51.3 \pm 0.2$ | 43.3 | $5.8 \pm 0.2$ | 3.9 | $7.6 \pm 0.3$ | 33.6 |
| **S-WMD INIT.** | $2.8 \pm 0.3$ | $28.2 \pm 0.4$ | $39.8 \pm 0.4$ | 38.0 | $3.3 \pm 0.3$ | 3.5 | $\mathbf{5.8 \pm 0.2}$ | 28.4 |
| **S-WMD (A ONLY)** | $2.5 \pm 0.4$ | $28.0 \pm 0.2$ | $39.7 \pm 0.3$ | 37.4 | $\mathbf{3.2 \pm 0.2}$ | 3.3 | $6.0 \pm 0.2$ | 27.2 |
| **S-WMD (w ONLY)** | $2.3 \pm 0.6$ | $27.6 \pm 0.2$ | $39.3 \pm 0.5$ | 37.6 | $\mathbf{3.2 \pm 0.2}$ | 3.6 | $6.1 \pm 0.1$ | 27.8 |
| **S-WMD** | $\mathbf{2.1 \pm 0.5}$ | $\mathbf{27.5 \pm 0.5}$ | $\mathbf{39.2 \pm 0.3}$ | **34.3** | $\mathbf{3.2 \pm 0.2}$ | **3.2** | $\mathbf{5.8 \pm 0.1}$ | **26.8** |

## 2 The top upweighted words on 20NEWS by S-WMD

It is interesting to see whether S-WMD can really identify most distinguishable words for classification tasks by learning the word weight **w**. We select the top 100 largest weights learned by S-WMD on the 20NEWS dataset, and find their corresponding words from the dictionary. The results are shown in Table 2. We also show the true classes of the 20NEWS dataset in Table 3.

Table 2: The top 100 words with largest weights learned by S-WMD on 20NEWS.

| Word | Weight | Word | Weight | Word | Weight | Word | Weight | Word | Weight | Word | Weight |
|---|---|---|---|---|---|---|---|---|---|---|---|
| windows | 4.03 | computer | 1.53 | ride | 1.31 | orbit | 1.23 | chip | 1.18 |  |  |
| sale | 3.44 | alt | 1.52 | copy | 1.29 | asked | 1.22 | saint | 1.18 |  |  |
| bike | 2.88 | atheism | 1.50 | rider | 1.29 | western | 1.22 | sun | 1.18 |  |  |
| mac | 2.73 | firearms | 1.50 | bus | 1.29 | Turkish | 1.22 | homosexuals | 1.17 |  |  |
| apple | 2.61 | motorcycles | 1.43 | Islamic | 1.28 | autos | 1.21 | DOS | 1.17 |  |  |
| gun | 2.41 | Israeli | 1.43 | auto | 1.28 | cute | 1.21 | flight | 1.17 |  |  |
| space | 2.30 | RISC | 1.42 | offer | 1.27 | crypto | 1.21 | SCSI | 1.17 |  |  |
| car | 2.25 | Israel | 1.39 | key | 1.27 | polygon | 1.21 | biblical | 1.17 |  |  |
| DOD | 2.23 | Rutgers | 1.37 | biker | 1.26 | playoff | 1.20 | powerbook | 1.17 |  |  |
| graphics | 1.90 | motif | 1.36 | animation | 1.26 | fired | 1.20 | NASA | 1.17 |  |  |
| guns | 1.87 | sell | 1.36 | monitor | 1.25 | dolphins | 1.20 | government | 1.16 |  |  |
| hockey | 1.86 | motorcycle | 1.36 | rocket | 1.25 | label | 1.20 | compatible | 1.16 |  |  |
| bikes | 1.79 | drivers | 1.35 | talk | 1.25 | security | 1.19 | happening | 1.16 |  |  |
| baseball | 1.76 | automotive | 1.34 | image | 1.25 | virtual | 1.19 | hell | 1.16 |  |  |
| driver | 1.67 | lists | 1.34 | Boone | 1.25 | Keith | 1.19 | motherboard | 1.16 |  |  |
| encryption | 1.64 | circuit | 1.33 | summarized | 1.24 | forsale | 1.19 | tapped | 1.16 |  |  |
| card | 1.63 | moon | 1.33 | riding | 1.24 | fit | 1.19 | TIFF | 1.15 |  |  |
| pro | 1.62 | Armenian | 1.33 | warning | 1.23 | diamond | 1.19 | mouse | 1.15 |  |  |
| story | 1.58 | NHL | 1.32 | controller | 1.23 | IDE | 1.18 | Mazda | 1.15 |  |  |
| clipper | 1.57 | shipping | 1.31 | doctor | 1.23 | electronics | 1.18 | gay | 1.15 |  |  |

Table 3: The class labels of 20NEWS

| comp.graphics | rec.autos | sci.electronics | talk.politics.guns |
|---|---|---|---|
| comp.os.ms-windows.misc | rec.motorcycles | sci.medsci.space | talk.politics.mideast |
| comp.sys.ibm.pc.hardware | rec.sport.baseball | sci.space | talk.religion.misc |
| comp.sys.mac.hardware | rec.sport.hockey | misc.forsale | alt.atheism |
| comp.windows.x | sci.crypt | talk.politics.misc | soc.religion.christian |

By comparing the top ranked words and the true class labels, we can observe that S-WMD indeed finds the most representative words for the classes. Since the weight vector **w** controls the contribution of each word to pairwise document distance, the model should assign high weights to words that are indicative of the class. For example, the word "windows" most likely corresponds to the class *comp.os.ms-windows.misc* or *comp.windows.x*; the word "sale" can be most helpful for identifying documents from the class *misc.forsale*; the word "bike" may frequently appear in the class *rec.motorcycles*; and the word "mac" is most representative for the class *comp.sys.mac.hardware*.