[Reviews · NeurIPS 2016]

Reviewer 1

Summary

This paper proposes a supervised metric learning approach built on top of the recent “word-movers distance” [19] (a transport distance applied to word2vec word embeddings). The proposed approach can be applied to any transport distance based on squared Euclidian distances after linear projection. On top of learning the linear projection to maximise performance, the proposed approach also learns a weighting of words. The latter is specific to the document application, it consists in re-weighting the frequency of each word by a learned weight, and re-normalising the word frequencies. The paper presents an efficient optimisation method to learn the proposed parameterised metric. A thorough experimental evaluation shows the effectiveness of the proposed approach by comparing it to a large number of reasonable baselines on a number of document classification datasets using a kNN classifier based on the proposed learned metric.

Qualitative Assessment

+ This work extends the recent word-movers distance, which is an interesting technically and useful extension (shown to be effective on a number of kNN text classification tasks). - The potential impact is perhaps limited by the fact that earth-movers distance with a metric learning via a “word centroid distance” initialisation performs similar to the full learning of the proposed distance. See the difference of the last two rows in Table 3. In other words, the main technical contributions of this paper do not seem to contribute very much over this initialisation method based on existing techniques. - One thing that I missed in table 3 is an evaluation of the “word centroid embedding” (weighted mean of word embeddings for the document) with all tested existing metric learning approaches (ITML, LMNN, NCA). - Another point missing in the evaluation is an assessment of the relative importance of the word-specific weights, and the projection matrix A. It would be useful to see performance with each one of these components being active in isolation. o The originality is mostly related to the way to optimise the metric (Line 156–175), the idea of a linear metric learning within the earth-movers distance is very interesting but a limited contribution in itself. + The paper is very well written and a pleasure to read. There are a few minor issues that could be clarified: - Line 152: “The authors”: of which paper? [1,7,8] ? - Equation 11: shouldn’t alpha and beta carry stars here ? - It would be useful to explicitly recall around line 170 that this is the Sinkhorn scaling that is referred to in the introduction in line 56.

Confidence in this Review

2-Confident (read it all; understood it all reasonably well)


Reviewer 2

Summary

An extension to the word mover's distance which uses supervision to learn weights on the distance. The paper features a very extensive experimental evaluation.

Qualitative Assessment

I am happy to see that someone picked up the approach of word mover's distance and used discriminative training to learn a domain-specific metric. The method is sound, well explained, with just the right information. The experiments are very extensive on several classification data sets and a wide variety of baseline methods and features. The suggested method outperforms all by a large margin. This is a solid paper that I would very much like to see presented at NIPS.

Confidence in this Review

2-Confident (read it all; understood it all reasonably well)


Reviewer 3

Summary

The authors propose a new method to learn the parameters of the WMD, basically the EMD computed on text documents using word embeddings to compute bin-to-bin dissimilarities. The authors parameterize the EMD cost (i.e. ground metric) matrix using a Mahalanobis distance between vector representations. The authors propose also to learn in addition to this some histogram coefficients reweighting (see Lebanon's work, and comment below). The authors provide the gradients of NCA type costs for this problem, and provide compelling experimental evidence.

Qualitative Assessment

Overall the paper reads like a nice combination of existing tricks, and provides very convincing experimental results. I would support acceptance. Strengths of the paper are simplicity and a relatively straightforward idea, but not trivial to implement/test. The experimental section is therefore a strong part of this paper. Things to improve: handle better the interplay between regularized/not regularized formulations, be more rigorous with maths (computations/notations are a bit sloppy) and ideally provide an algorithmic box to see more clearly into what the authors propose. A few minor comments: - In Eq.1, the Euclidean distance between word embeddings is used as a cost, in Eq.6, for the purpose of Malahanobis metric learning, that cost becomes the squared euclidean metric (and thus what is usually referred to as 2-Wasserstein). To avoid such inconsistencies, it's probably easier to simply say that WMD is p-Wasserstein between clouds of word embeddings, whatever the p. - The "reweighting" approach proposed in Eq. 5 is not new, it was already proposed by Guy Lebanon in exactly the context of text document classification ("metric learning for text documents"). See also "Unsupervised Riemannian Metric Learning for Histograms Using Aitchison Transformations" by Le/Cuturi. - Any experimental insights on the importance of the weight vector w in the learned metric? does that innovation matter? is it relevant in any way? - Related work: it would be interesting to underline more clearly differences with previous work. For instance, this paper builds clearly on the work of Cuturi/Avis, but uses entropy regularization, which clearly speeds up / scales up applications. The cost is also different: while discrete in that work, it's now continuous but parameterized with linear maps. These two arguments are missing. This paper also draws from Wang et al, but that paper is not publishable as it is. The OT optimization is also run once only in their paper, and clearly does not work as it was proposed.

Confidence in this Review

3-Expert (read the paper in detail, know the area, quite certain of my opinion)


Reviewer 4

Summary

This paper shows a supervised way of learning the distance between documents on top of word mover’s distance (WMD). WMD is proposed in a previous literature motivated from earth mover’s distance. More specifically, from one distribution of words to another distribution, optimal transport is calculated to obtain the document distance. For the transport distance between words, word2vec embedding is initially used. In this paper, the authors propose learning (1) a linear transformation of embedded words and (2) a reweighted word distribution, so the documents of different labels are more separated.

Qualitative Assessment

This paper is clearly written and easy to follow. The paper contains a thorough review of previously related works and the authors have a good understanding of those literatures. The authors provide experimental results of their algorithm for many novel datasets. However, the motivation of the formulation needs more explanation, and the significance is low. 1) Why the authors use reweighting of the word distribution using w? This is not well-motivated, and it seems this reweighting can change the document into a completely different document. If the authors have shown why this reweighting helps without changing the document into a different document, then this reweighting could be justified. 2) The contribution of this work is not significant. Adopting linear transformation `A’ and reweighting `w’ with NCA objective function is one possible extension for supervised setting of WMD, but this formulation is not groundbreaking nor fancy. 3) How this algorithm can find a distance for a new document? For a completely new document, how one can find a reweight w? 4) It seems `the semantic difference’ in the abstract indicates the label of document. Authorship difference or topic difference does not seem a semantic difference. Does the semantic difference mean something else? In general, the paper is nicely written, and I do not object the acceptance of this paper once the questions are appropriately explained.

Confidence in this Review

2-Confident (read it all; understood it all reasonably well)


Reviewer 5

Summary

The authors analyze the problem of text classification into categories. They expand a previous work that suggested Word Mover's Distance (Wasserstein) to make it supervised by learning word specific weights. This contribution suggests to use the Sinkhorn distance as proxy for the Wasserstein distance to improve its speed, and evaluates on 8 datasets against 26 baselines showing the best average performance.

Qualitative Assessment

This is a very well written paper that presents well the topic and analyzes the math behind the problem in depth but also with great clarity. The only weaknesses are, maybe, the limited novelty compared to the previous Word Mover's Distance paper, and I don't find the references against the World Centroid Distance relevant, but confusing. I have the feeling that lambda (the regularizing factor) is poorly discussed. In my experience the value of lambda has a very large impact, sometimes being the dominant factor of the eq(2). Theoretically, results should be similar to pure Wasserstein distance given a large enough lambda at the cost of slower convergence, but most people using Sinkhorn use a very small lambda to improve converge rate, achieving a behavior plainly different from Wasserstein. So I am curious if, in this contribution, lambda dictates a pure Wasserstein behavior or a Sinhorn (entropic) behavior.

Confidence in this Review

2-Confident (read it all; understood it all reasonably well)


Reviewer 6

Summary

The paper proposes a new method for metric learning for textual documents, building upon previous work known as "Word Mover's Distance". The original is unsupervised, measuring the distance between the embeddings of words. This paper extends this to the supervised setting, where the supervision is of the type 'these documents belong to the same class'. Code (matlab) is promised to be made available.

Qualitative Assessment

[I appreciated the careful responses of the authors which clarify my questions] The authors propose two things to learn the distance: one is a linear transformation A (the traditional approach), and the other is the word importance weights w. I would have liked to understand better the intuition behind Eq (5) (which defines the usage of w), and understanding which of both parts is responsible for the reported improvements. In the experimental sections, the method is evaluated in a classification task. I understand that evaluating a distance metric is not easy, but - just for comparison - it would be interesting to see how well a simple classifier does on the same datasets. [these are now irrelevant, as clarified by the authors] Questions: I did not understand when Eq 2 is used.The lambda there seems to be different than the lambda in ln170 (?) What are your arguments for the claim "which is particularly forgiving when two documents use syntactically different but conceptually similar words" (ln316)? Other comments: ln257: "on the right we also show the average error across datasets". I couldn't find this, is this supposed to be in Table 3? Table 2: what is the difference between "n" and "ne"? I couldn't find mention on how k (for kNN classification) is defined. Is this data-set or method dependent?

Confidence in this Review

1-Less confident (might not have understood significant parts)